# Establishing a robust testing approach for displacement measurement on a rotating horizontal axis wind turbine

Nadia Najafi[1], Allan Vesth[1]

[1]Wind Energy Department, Technical University of Denmark, Denmark

*Correspondence to*: Nadia Najafi (najafi.nadia@gmail.com)

**Abstract.** Health monitoring by conventional sensors like accelerometers or strain gauges becomes challenging for large rotating structures due to the issues with feasibility, sensing and data transmission. In addition acceleration measurements have low capability in presenting very small frequencies which happen very often for large structures (for instance frequencies between 0.2 and 0.5 Hz in horizontal axis wind turbines). By contrast displacement measurement using stereo

vision is rapid, non-contacting and also distributed over the structure. The sensors are cheaper and easier to be applied in many places on the object to be measured. Horizontal axis wind turbines are one of the important large rotating structures which need to be measured and monitored in time to prevent damage and failure, and the blade tip position is one of the key parameters to measure in order to prevent blade hitting the turbine tower.

This paper presents a clearly described and easily applicable procedure for measuring the displacement on the components of

a rotating horizontal axis wind turbine (HAWT) with stereophotogrammetry. Paper markers have been applied on the rotor and tower of a scaled down HAWT model in the workshop and the displacement measurement method has been demonstrated by measuring displacement during operation. The method is mainly developed in two parts: 1) camera calibration and 2) tracking algorithm. We introduce an efficient camera calibration method for measurement in large fields of view that has always been a challenge. This method is easy and practical and offers better accuracy compared with 2D

traditional camera calibration. The tracking algorithm also works successfully and is able to track the points during rotation within the measurement time. Finally the accuracy analysis has been conducted and has shown better accuracy of the new calibration method compared with 2D traditional camera calibration.

## 1 Introduction

Regarding the increasing industrial advances and the world's population growth, fossil fuel sources will not meet the human

need for energy in near future. In such a situation wind energy is an environmental friendly alternative that can decrease the dependency on the declining fossil fuels (Herbert et al., 2007). Prospering the wind turbine technology has led to important concerns about the reliability of the energy production and wind turbine efficiency and reliable turbine operation also requires proper experimental tools and high quality testing methods to monitor the turbine behaviour.

Traditional contact transducers such as strain gauges and accelerometers have been used for vibration analysis, health monitoring, damage detection and structural displacement of wind turbines (Weijtjens et al., 2017; Yang et al., 2014; Osgood et al., 2010; Lorenzo et al., 2016; and Manzato et al., 2014) and other large structures like bridges (Hoffmann, 1989; Fukuda et al., 2013; Park et al., 2005; Ye et al., 2012; Xia et al., 2014 and Siriwardane, 2015), but they have difficulties in measuring in large scale; the installation process that often includes wiring is costly and time consuming. The measured signal from conventional sensors such as accelerometers is not very accurate in measuring low frequencies of the structure (for instance frequencies between 0.2 and 0.5 Hz in horizontal axis wind turbines) and includes the centrifugal components (Najafi and Paulsen, 2017). In addition, contact transducers can only measure the structure in a few numbers of points and increasing the measurement points creates additional cost and complication (Hunt, 1998). As an alternative, non-contact optical measurement techniques provide faster and cheaper possibility to measure displacement on large and rotating structures such as wind turbines. Stereophotogrammetry is one of the common optical techniques for motion tracking of the objects that enables 3D displacement measurements. Stereophotogrammetry or stereo vision estimates the 3D coordinate of the points using two or more 2D images taken from different angles. The preparation time is short and it could measure at many points on large structures.

Displacement of the turbine components (blades and tower) is an important parameter in analysing the rotor performance and structural behaviour of the turbine during operation. Stereophotogrammetry has been previously employed to estimate the strain and full displacement field of the turbine blades with digital image correlation (DIC) for investigation of the relative out of plane blade deflections (Winstroth, et al., 2014), rotor vibration measurement (Waren, et al., 2010 and Poozesh, et al., 2016) and blades damage detection (Leblanc, et al., 2013 and Zarouchas and Hemelrijck, 2014). DIC gives the continuous displacement distribution and is computationally expensive in monitoring large scale structures, but 3D point tracking (3DPT) which measures displacement in discrete points, is the preferable approach for outdoor and large scale experiments. In 3DPT, the optical targets, that can be simple paper markers (reflective or not reflective), are mounted at different places on the structure, as many as desired, and their 3D coordinates are tracked in time. 3DPT has been used for displacement measurements of the turbines in the recent years. The displacement measurement via 3DPT has been used to predict the full field dynamic strain of a model scale wind turbine blade (Baqersad, et al., 2015). However, stereo vision is new in measuring vibration; it shows good agreement with conventional transducers like accelerometers and strain gauges in this field (Warren, et al., 2010b and Najafi, et al., 2015). In, Najafi and Paulsen, 2017, 3DPT has been used to study structural vibrations of a model scale vertical axis wind turbine. Najafi and Paulsen, 2017, have investigated the challenges of using stereo vision for vibration analysis of complex geometries with sharp curvatures and out of plane components. In (Prowell, et al., 2011; Prowell, et al., 2012 and Paulsen, et al., 2012) the displacement measurements by point tracking stereophotogrammetry is used for determining structural response and modal properties of utility-scale horizontal axis wind turbines.

The current study is focused on establishing a clearly described and easily applicable procedure to measure displacement on the components of a rotating horizontal axis wind turbine using stereo vision technique. A scaled down HAWT model is

used to demonstrate the displacement measurement method. Camera calibration is one of the main challenges in measurement in large fields of view, which links the 3D coordinate of the points in the real world to their corresponding 2D coordinate in the image plane. Traditional way of calibration uses a calibration object with known and precise coordinates to calibrate the camera. This method is accurate and also efficient but it is unpractical for large field of view applications

because of the calibration object size. There are also other techniques of calibration with no need for calibration object, which are called self-calibration. Self-calibration uses epipolar geometry of stereo pairs to reconstruct the 3D coordinates. These methods are flexible but the final results are not always precise and reliable because there are many parameters that need to be estimated (Medioni and Kang, 2005). In this study we updated the traditional calibration method for large fields of view to be easier, faster and more practical. We also compared the results of the updated 3D calibration method with the

traditional calibration that is conducted with a large grid in the background and the comparison shows better accuracy of the new 3D calibration procedure.

## 2 Experimental setup

The case study is a scaled down model of 3.6 MW turbine (see Figure 1). The rotor diameter is 1640 mm and the blades are made of Aluminium with rectangular cross section of 5 mm×8 mm. The tower is also an Aluminium rod with the height of

1600 mm and the cross section of 16 mm in diameter. The rotational speed of rotor is between 0-150 rpm. A wire is twisted around the individual blades to prevent vibrations induced by vortex that has formed behind the rectangular shape of the blade.

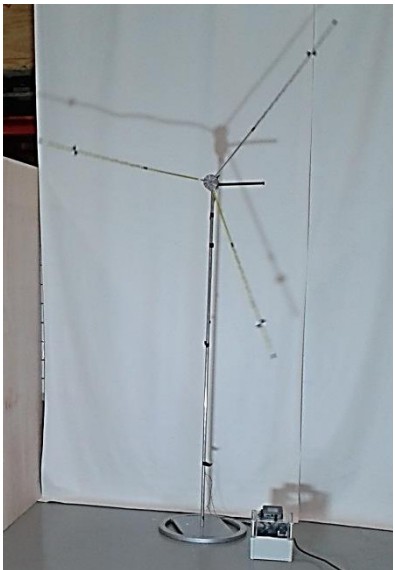

**Figure 1:** scaled down model turbine

Stereo vision measurement system contains two Basler acA2040-180km cameras equipped with 40 mm (focal length) Nikon lenses. The maximum frame rate of the cameras is 187 frames per second (fps) for full size images. The image acquisition system has limited capacity; therefore the longest acquisition time of stereo system with maximum frame rate (187 fps) is about 16 seconds. However with decreasing the frame rate the acquisition time increases, for instance the motion of the turbine can be tracked for about 150 seconds with 20 fps. Nevertheless with upgrading the acquisition system the measurement time can be modified.

In this experiment the cameras are placed about 7.5 m away from the turbine while they are apart by about 3.5 m. According to the turbine dimensions the imaging area is chosen to be about 2 m×2 m. Based on the focal length, camera sensor size, and assuming that the cameras follow the pinhole camera model, the distance between the cameras and turbine is calculated to be 7.5 m.

This setup satisfies the rule of thumb which says that the distance between the cameras should be at least 1/3 of the distance between the cameras and the test object; it can be up to 3 times the distance as long as all the targets on the object can be seen in the stereo image pairs.

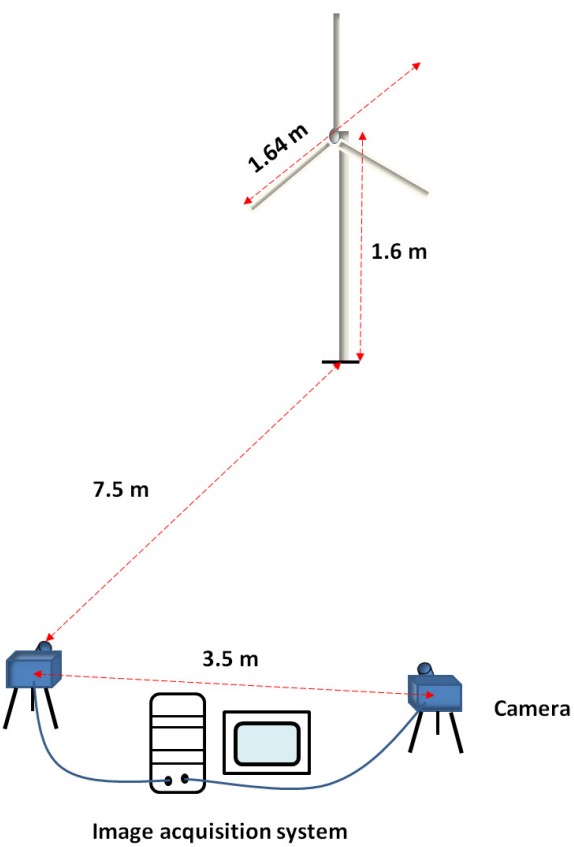

**Figure 2:** Stereo vision system set up

The simplest marker used in stereo vision is circular with a good contrast with the background and the center of the marker is considered to be tracked. According to (Ozbek and Rixen, 2013), curvatures of the structure where the markers are applied, lead to perspective errors that need to be corrected. In other words, in horizontal axis wind turbines, some parts of the blades are curved or deformed due to the loading during rotation, this deformation and also the relative angles between camera and turbine cause the changes in the shapes of the marker in the image from circular to elliptical. Therefor the real center of the circular marker should be calculated by knowing the exact relative angles and blade deformation at the marker position that is quite challenging due to the unknown instant blade deformation. To avoid these difficulties, we have changed the marker shape to the following shape, shown in Figure 3, with a diameter of 4 cm. In this case there is no need to calculate the center of the marker and it can be found using robust corner detection image processing algorithms independent from the marker shape in the image and blade curvatures.

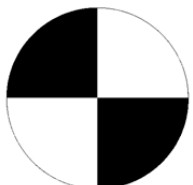

**Figure 3:** Marker shape

**3 Camera Calibration**

Camera calibration is the estimation of intrinsic and extrinsic parameters of the cameras in order to relate the 3D coordinates of the real word to 2D coordinates of the image. Extrinsic parameters define the location and orientation of the camera that contain the translation vector and the rotation matrix. Intrinsic parameters describe the optical, geometrical and digital characteristics of the camera such as focal length, image centre in pixel coordinates, the effective pixel size in horizontal and vertical directions, and the radial distortion coefficient. Traditional camera calibration is the most common way of calibrating cameras and has been studied and improved during years by different researchers (Tsai, 1987; Weng, et al., 1992 and Zhang, 2000). This method uses a calibration object including a number of points with known coordinate to estimate the camera parameters. The traditional calibration is divided to three methods based on the calibration object dimensions: 3D, 2D (planar) and 1D (linear) calibration (Medioni and Kang, 2005). 3D calibration can be conducted very efficiently with very high precision but it requires expensive equipment and elaborate setup in the traditional calibration procedure (Faugeras, 1993), however 2D traditional calibration is easier and less expensive (Sun and Cooperstock, 2005).

In this section two calibration methods are applied: at first the cameras are calibrated using a 2D calibration board with known coordinates and in the second part a new 3D calibration approach is introduced.

### 3.1 2D traditional camera calibration

The 2D traditional calibration has been conducted with a 2D printed grid applied on a wooden board. The dimensions of the grid are 2 m×1.9 m and it contains 21 horizontal and 20 vertical lines that intersect at 420 points with known coordinates that are 10 cm apart in horizontal or vertical direction.

The exact positions of the grid point in each image are determined using image processing algorithms:

Firstly a part of the grid that is common between the fields of view of both cameras is chosen, and then 8 points on the borders of that area are picked (see Figure 4: Left). A fraction of the image around each of the border points is taken (image window) and the exact position of the points is given by intensity median of the image window.

     The coordinates of windows around all the grid point are estimated using interpolation between the border points and

finally the exact positions of the grid points in each image are determined as the intensity median of the windows (Figure 4: Right).

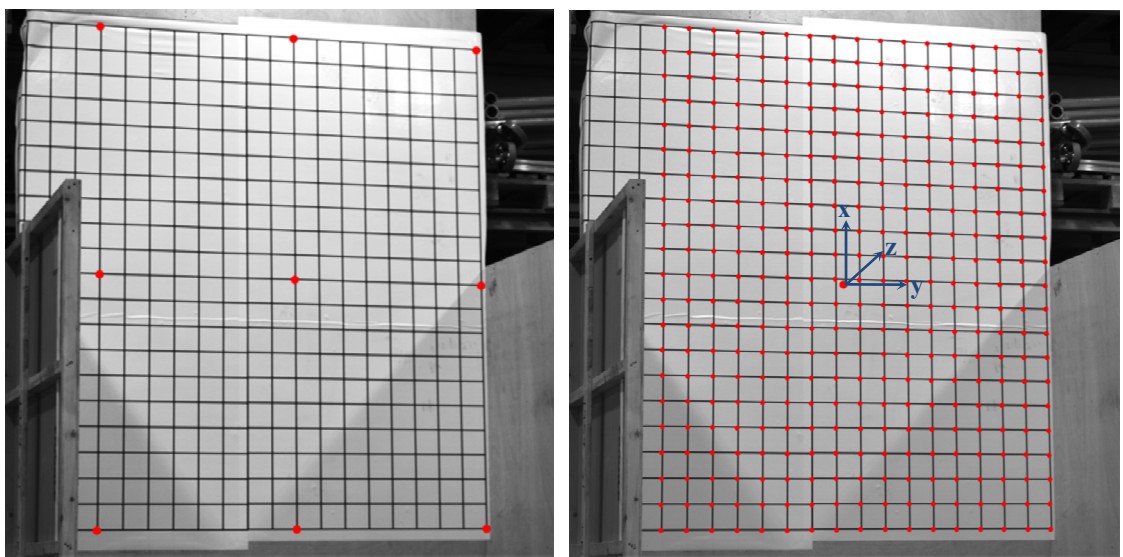

**Figure 4: Left:** 8 points on the borders of the common between the fields of view of both cameras on the calibration grid, **Right:** Identifying the calibration points in calibration grid

After getting the exact pixel positions of the grid points in the image and also having the coordinates of the points in the real world, the traditional calibration builds the equations that relate the coordinates of the real world to the coordinates of the images taken by the camera. And finally intrinsic parameters of the camera (focal length, chip size, image center, etc.) and extrinsic parameters (translation and rotation matrix) are defined.

### 3.2 3D updated camera calibration by a surveillance device with a laser range finder

Traditional 3D calibration method can be conducted very accurately, as it has been discussed in the beginning of this section, but it needs expensive and elaborate setup and expensive equipment such as two or three orthogonal planes. Setting up the

traditional 3D calibration in large fields of view, like full scale wind turbine experiments, is not feasible due to the difficulties in providing a precise calibration object. In our new calibration method, instead of using a huge 3D calibration object, we used the markers that are applied on the turbine for the sake of measurement, as the calibration points. A Leica surveillance device (Leica Nova MS50) determines the exact position of the points quite accurately and then the coordinates are used for 3D calibration of each camera. Leica Nova MS50, is a surveillance device with a laser range finder that uses advanced technologies for 3D laser scanning, imaging and GNSS (Global Navigation Satellite System) positioning. The accuracy in $x$ and $y$ directions is dependent on the distance between the Leica Nova MS50 and the object, hence the accuracy in $x$ and $y$ directions in the current case is about 0.0349 mm. In addition the Leica Nova MS50 accuracy in depth ($z$ direction) is 1 mm for measuring on reflective surfaces.

During the calibration the rotor was rotated (manually) by a specific angle step by step, in order to cover the whole rotor area by the calibration points and establish the collection of known coordinates for camera calibration.

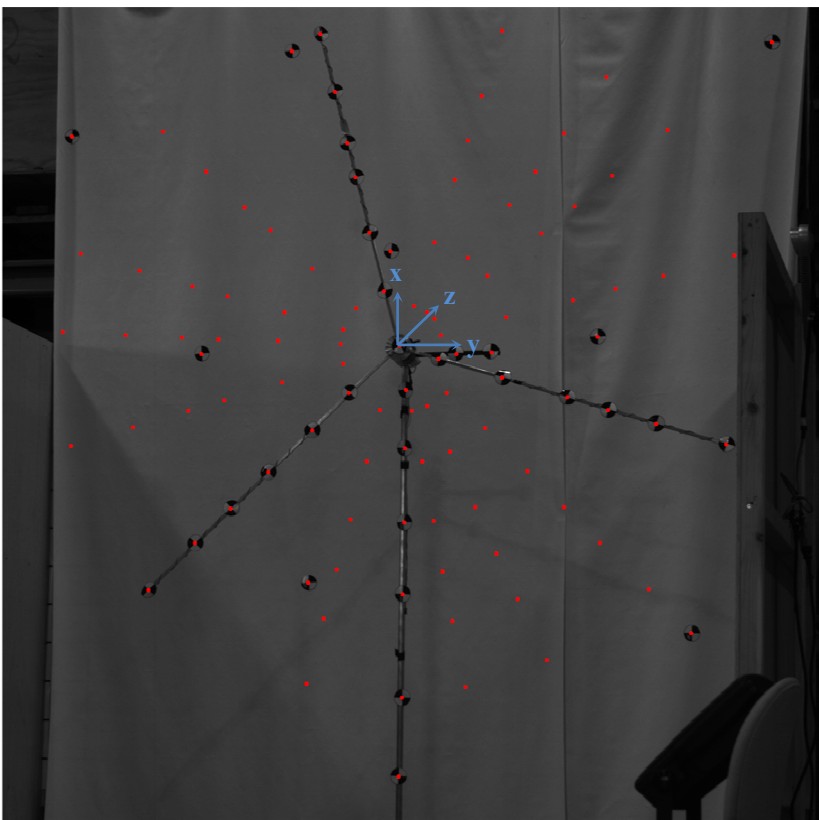

**Figure 5:** calibration points in the new method. The blue axes show the real world coordinate system

## 4. Tracking algorithm

To measure the displacement of the markers during the rotation, a tracking algorithm has been developed. The main steps of the algorithm are:

1- Picking the coordinates of the markers in a number of image sequences during a cycle of rotation with equal intervals between the sequences. For example if the camera takes 40 pictures during one rotor revolution, we picked the coordinate of the markers in every 4 pictures to estimate the path of each marker. In this step an ellipse equation is fitted to the markers path during one rotation.

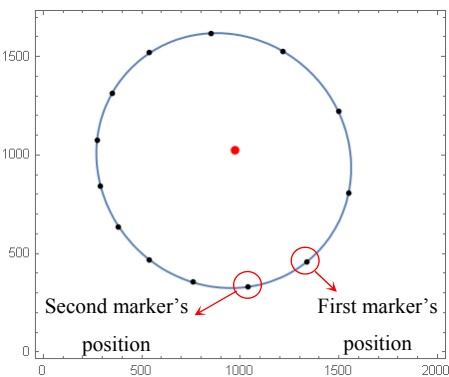

**Figure 6:** Elliptical path of a marker in the image coordinates during rotation, **Black dots**: the coordinate of the marker that has been picked with equal intervals, it means there are N1 coordinates of the marker between every two black dots, **Blue line**: the fitted ellipse, **Red dot**: the center of the ellipse

2- **Initial guess for angular deflection between image sequences:** The angle between the lines that connect the ellipse center to the first and second picked marker coordinates on the ellipse in step 1 (black dots in Figure 6) is calculated, and by knowing the number of image sequences between the first and second positions of the marker (N1) on the ellipse, the first guess for the angular deflection between image sequences is obtained. As the turbine rotational speed is not fully constant during rotation, we need to update the angular deflection in each sequence.

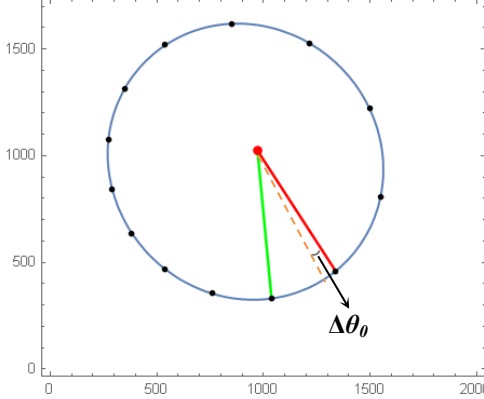

**Figure 7: Red**: The line between the elliptical path center and the first position of the marker at time $t=t_0$, **Green**: the line between the center of the elliptical path and the position of the marker after a number of sequences

3- With the initial estimation of the angular deflection between the image sequences ($\Delta\theta_0$) and also the elliptical path of each marker, the approximate position of the marker in the next sequence (time of $t_0+\Delta t$) is estimated.

4- An image window is established around the approximate position of the marker and the exact position of the marker is calculated using the Harris-Stephens corner detection algorithm in the window. This algorithm is described in Harris and Stephens, 1988.

5- By knowing the exact position of the marker, the exact angular deflection is calculated ($\Delta\theta$).

6- The approximate marker position in the next sequence is estimated using the angular deflection in the previous sequence. On the other hand, the initial guess for the angular deflection in each time step is the angular deflection of the previous sequence ($\Delta\theta_0 = \Delta\theta$).

7- The exact position of the marker is estimated by repeating the algorithm from step 3 to 5.

The flowchart in Figure 8 briefly explains the tracking algorithm.

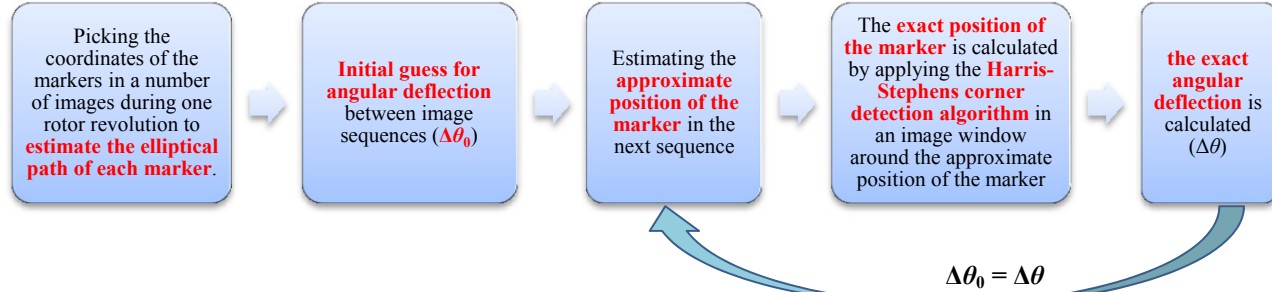

**Figure 8:** Tracking algorithm flowchart

## 5. Results and discussion

After calibrating the cameras by traditional and updated calibration methods, the 2D positions of the markers in the image are found during the rotation using the tracking algorithm. The markers that are followed in time are shown in Figure 9:

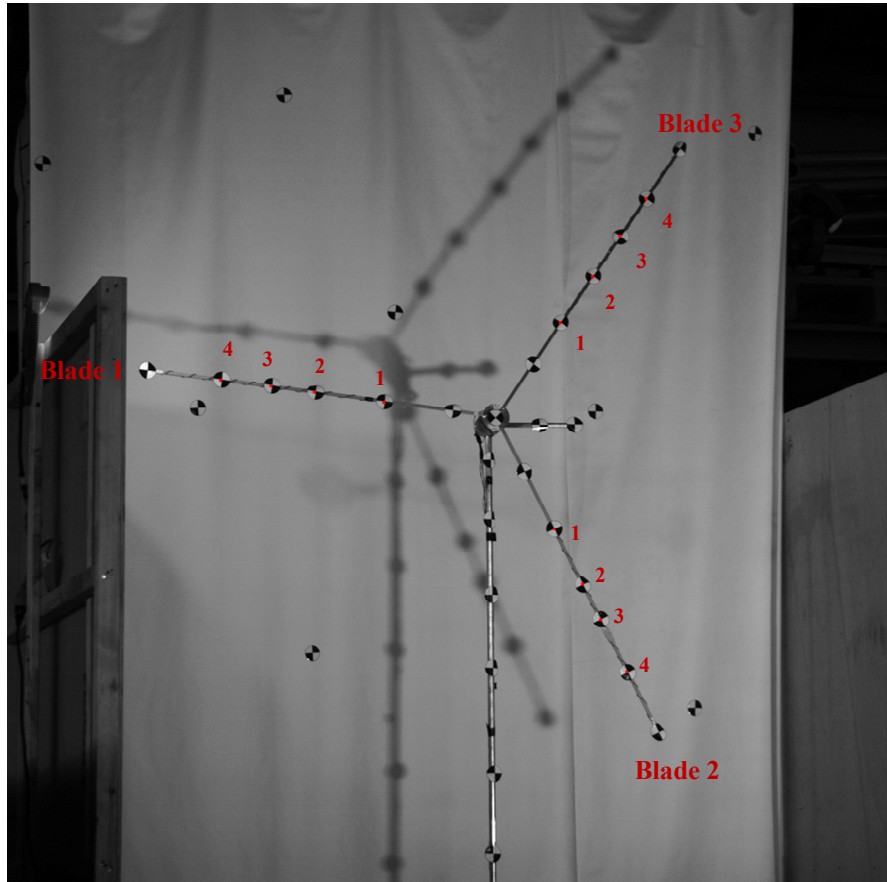

**Figure 9:** Marker numbering on the rotor

The line of sight from each marker to each of the cameras is calculated by both calibration methods and the intersection of the two lines of sight is found as the 3D position of the marker (stereo triangulation). The 3D updated calibration has been done with different numbers of calibration points to compare the results: 1) all of the 111 points including the points on the turbine rotor, tower and on the background, 2) 103 points including the points on the turbine rotor and tower, 3) 35 points of the points on the turbine rotor and tower.

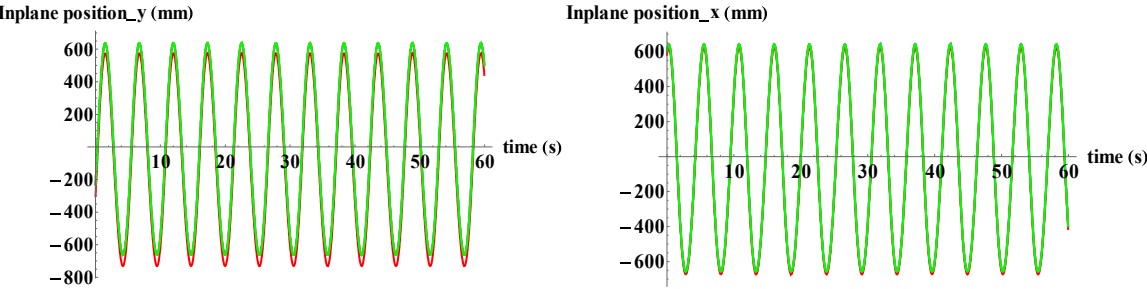

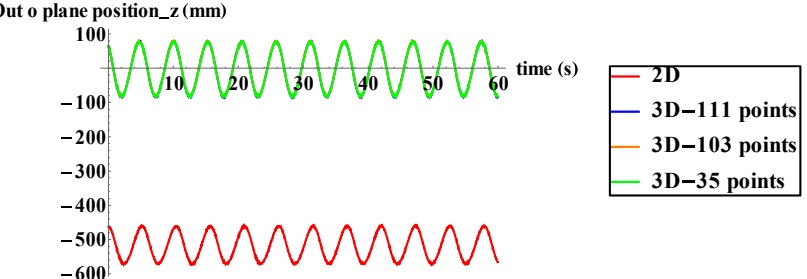

**Figure 10:** Displacement time series for point number 4 with different calibrations (plots corresponding to 3D calibration are almost on top of each other)

It can be seen in Figure 10 that there is an offset, especially in depth direction, between the displacement obtained by the traditional and updated calibration methods; this offset is due to having different origins of the coordinate systems in 2D and 3D calibrations. There are also other minor differences between traditional 2D and new 3D calibrations (is more obvious in in-plane positions) that a part of it can be due to the slightly different directions of the coordinate axes.

For comparing calibration methods two different indicators are investigated: 1- the distance between the lines of sight at the intersection position, 2- the distance between two markers during rotation.

The lines of sight from the marker to the cameras do not exactly intersect in the space due to the inaccuracies, and the 3D position is regarded as the point with minimum distance from two lines of sight (Trucco and Verri, 1998). Therefore the distance between the lines of sight is considered as an indication of measurement inaccuracy that is mainly caused by calibration uncertainties, light reflections and also by other environmental and physical factors.

The distance between the markers will barely change during the turbine operation due to the centrifugal forces and gravity, thus the change of the distance between the markers when the turbine is rotating can be used as another indication of the measurement inaccuracy.

In the following table the averaged and the maximum value of the distance between the lines of sight during rotation for point number 2 on blade 2 are listed for different calibrations and in different rotational speeds. Light and the other environmental factors are almost the same in all the measurements, therefore the different values of lines of sight distances are mostly due to different calibrations. The photography sampling frequency is 50 frames per seconds and the recording time is 1 minute.

**Table 1:** The distance between the lines of sight during rotation for point number 2 on blade 2 and for different calibrations

|  | 5 Hz | | 10 Hz | | 30 Hz | |
|---|---|---|---|---|---|---|
|  | **Mean** | **Max** | **Mean** | **Max** | **Mean** | **Max** |
| **2D- calibration grid** | 34.63 mm | 79.71 mm | 35.24 mm | 77.75 mm | 43.99 mm | 82.81 mm |
| **3D- 111 points** | 2.81 mm | 12.09 mm | 2.66 mm | 10.69 mm | 5.44 mm | 9.63 mm |
| **3D- 103 points** | 2.79 mm | 11.44 mm | 2.60 mm | 10.09 mm | 5.05 mm | 9.13 mm |
| **3D- 35 points** | 3.04 mm | 13.45 mm | 3.08 mm | 11.84 mm | 5.35 mm | 10.42 mm |

It is obvious in Table 1 that the distance between the lines of sight is much larger in the measurement with traditional 2D calibration compared to the measurement with the new 3D calibration. This comparison proves that the new calibration with Leica Nova MS50 is more accurate than the traditional 2D calibration, even with much fewer calibration points. The distances between the lines of sight in measurements with different numbers of the 3D points are relatively close, but the measurement of the 3D calibration with 103 points on the turbine shows slightly better result. It can be seen in Table 1 that by adding the background calibration points the distance between the lines of sight does not change significantly. This is a good and practically relevant point for full scale turbine measurement which shows that having other calibration points than the points on the turbine is not necessary and does not improve the displacement measurement quality. It is expected to see better results with more calibration points; however 3D calibration with one third of the turbine calibration points still looks acceptable. This effect of the number of the calibration points on the calibration quality is an important parameter that also needs to be checked in the full scale experiment.

According to (Ozbek and Rixen, 2013), distance between the target points for a real turbine remains constant during rotation. Therefore the blade elongation is studied for rotational speed of 5 rpm as an indicator for calibration precision. In Figure 11 change of the distance between markers 1 and 2 on blade 1 (shown in Figure 9) that are about 146 mm apart and markers 1 and 2 on blade 3 that are about 145 mm apart are plotted for the different calibrations approaches:

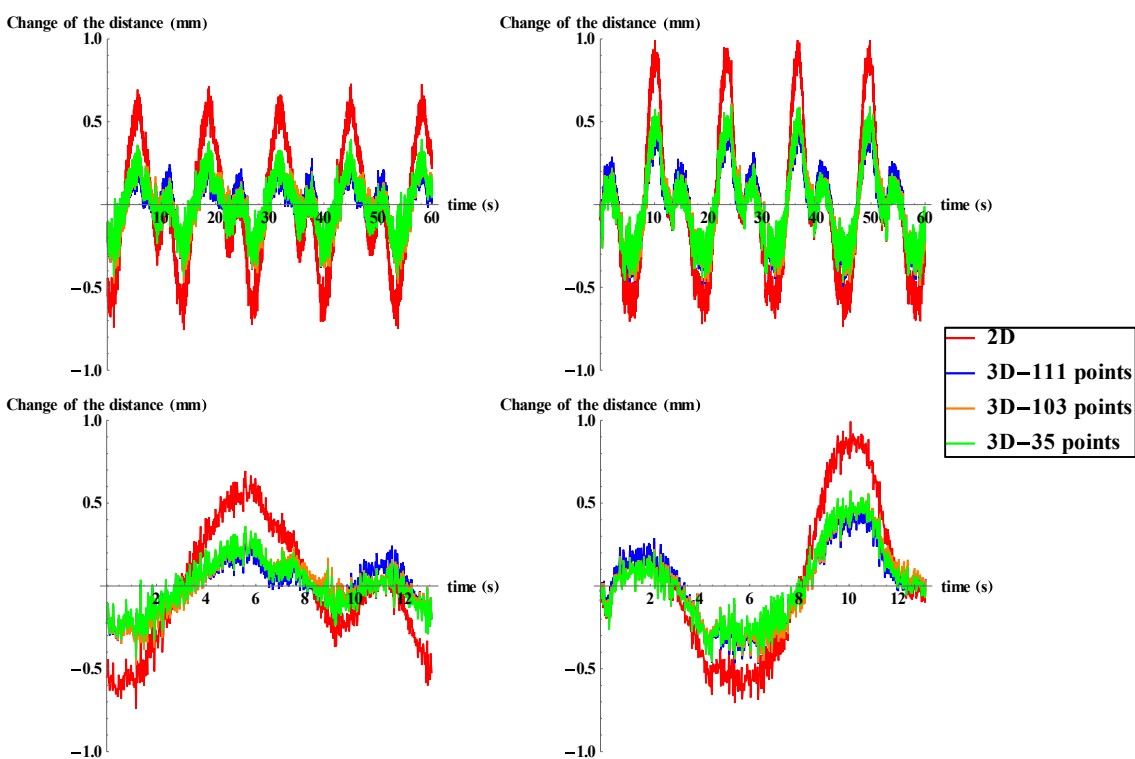

**Figure 11:** Elongation between markers 1 & 2 on blade 1(left), and blade 3 (right) in one minute (above) and during one cycle of rotation (bottom) with different calibrations

The larger change of distance between the markers with the 2D calibration can be correlated to the less accuracy in this calibration compared to the 3D calibrations with Leica surveillance device.

The distance between two markers changes periodically in Figure 11, no matter which calibration method is used to obtain the displacement. The distance between the other two markers at almost the same area of the blade (markers 1 and 2) are also checked for the sake of reliability and repeatability and they also revealed the periodic behaviour of the elongation between markers that is due to the repeating errors during operation. To investigate this behaviour, the spectrum of the distance measured with 3D calibration has been plotted in Figure 12.

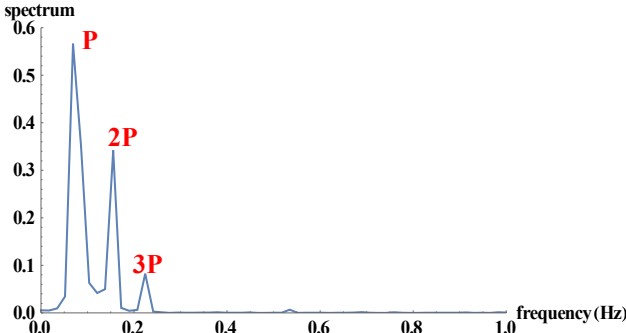

**Figure 12:** Power spectrum of the distance changes between points 1 and 2 on blade 3 calculated by 3D calibration using turbine points

Figure 12 shows that the periodic behaviour of the elongation between markers is dominant by P and 2P, where P is the rotor rotational frequency (5 rpm ≈ 0.083 Hz). This could be because of the physical problems such as the light reflection, calibration inaccuracies, geometry miscalculations and also camera un-synchronization.

By looking at the images that are taken and processed for the displacement measurements, it can be seen that each marker in Figure 11 is exposed to strong reflection once during one cycle that will be two times for a pair of markers, this could explain the 2P peak in the spectrum. For decreasing the unwanted reflections during displacement measurement, the markers can be printed with matt coating.

The geometry miscalculation which is mainly happening for circular markers is less likely to happen in the current study. On the other hands the image of circular markers changes to an ellipse due to the blade loading and deformation during rotation and relative angle between the camera and the turbine. To calculate the center of the circular marker from its elliptical image, knowledge about the exact angle of the rotor plane is required, but in the current case no center calculation is needed due to the specific shape of that marker (Figure 3).

In the current study there is no external trigger or switch to start the cameras and they are triggered at the same time using software trigger (LabVIEW code) that could disturb the perfect synchronization between the cameras. This also could be one of the sources that make the P peak occur in the spectrum. The effect of the camera un-synchronization would be obviously more pronounced in full scale experiments with very larger dimensions and higher rotational speeds; therefore the image acquisition system should be equipped with the external trigger during full scale turbine testing.

There are also other challenges that should be considered during the full scale operational turbine displacement measurement via the technique that is developed in this study. One of the main challenges is turbine vibrations due to the wind during the camera calibration that will not happen in the workshop. It is suggested that the calibration be done in a calm weather condition to minimize the inaccuracies of the coordinate measurement by Leica surveillance device due to the wind induced vibrations of the targets. Furthermore the wind induced vibrations should be considered in uncertainty analysis, however the Leica Nova MS50 coordinate measurement is based on wave form digitizing technology (WFD) that is able to send out the short pulses with a frequency of up to 2 MHz toward the target [Maar and Zogg] and it enables the Leica Nova MS50 to capture even very fast vibrations while the most dominant natural frequencies of the large wind turbines barley exceed 10 Hz. In addition the camera and Leica surveillance device should be synchronized during the calibration process to capture the changes of the turbine vibrations.

Another complexity about the field measurements is the changes of the wind speed and direction that lead to changes in turbine yaw and blade pitch angles. The calibration method that is proposed in this paper is a 3D calibration due to the 3D distribution of the points on the rotor, nacelle and turbine tower. Therefore if the turbine yaws, or the blades pitch within a reasonable range, the calibration is expected to be still valid for the measurement. However a comprehensive study should be done on the yaw and pitch angles range where camera calibration is valid in the field measurement.

The uncertainty of the measured displacement signal obtained with 103 points calibration is calculated using the generalized method based on the law of error propagation in a linear camera model of a stereo vision system. In this method, which is very well described in (Chen, et al., 2008), the uncertainty propagation in stereo reconstruction is explained in two main stages: camera calibrations and 3D triangulation to obtain the 3D coordinates from 2D projections in the images. In Figure 13 the mean value of the position determination uncertainty in $x$ ($U_x$), $y$ ($U_y$) and $z$ ($U_z$) directions is presented for markers shown in Figure 9.

(a)                                                        (b)

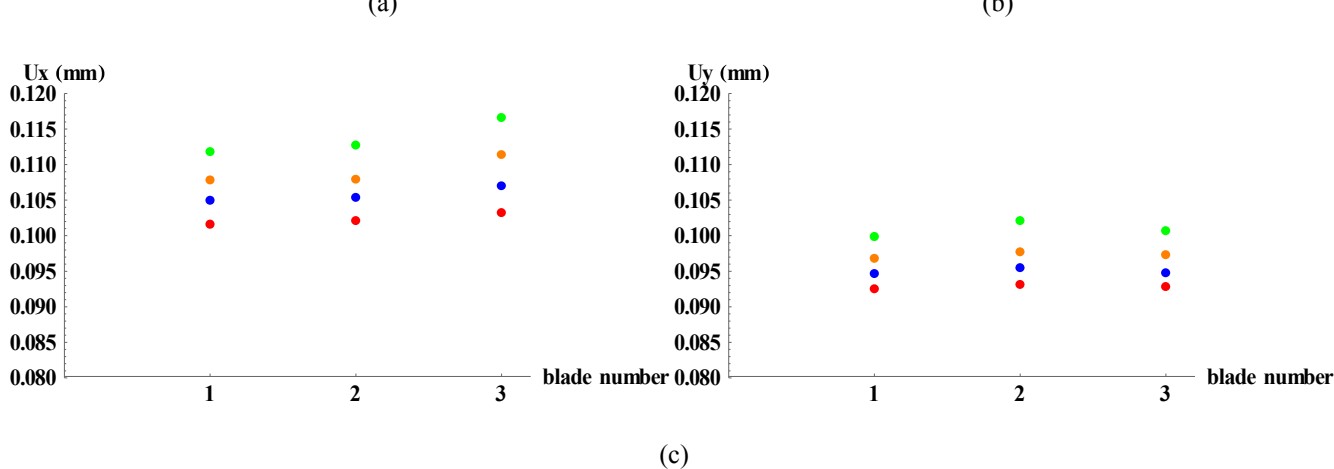

(c)

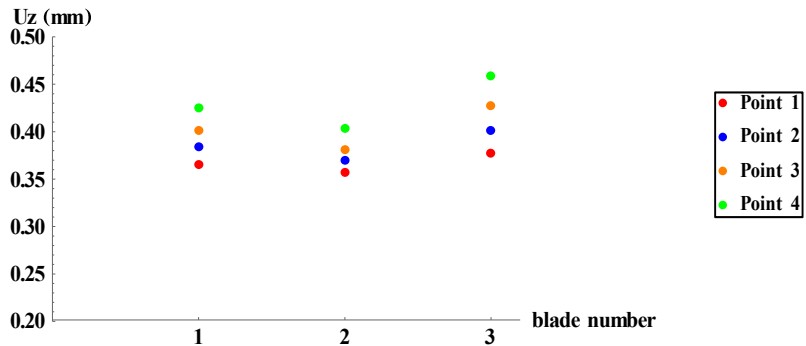

**Figure 13:** Mean value of the uncertainty for rotor points

It can be seen in Figure 13 that the uncertainty values do not change significantly from root to the tip of the blade which shows that the lens distortion is negligible.

## 6. Conclusion

In this paper, a clearly described and easily applicable stereophotogrammetry approach is introduced to measure the displacement of a rotating horizontal axis wind turbine. Camera calibration and marker tracking algorithm are studied in this work.

Camera calibration for large fields of view normally involves a lot of challenges. The traditional way of calibration that uses a calibration grid with known points is impractical for large scale uses. For example, with a modern size wind turbine the calibration grid would at least need to be $120 \times 120$ $m^2$, and other calibration techniques such as self-calibration are not always precise and reliable.

In this study a new 3D calibration method has been developed that is especially suitable for large fields of view; in the current case an operating horizontal axis wind turbine. The new 3D calibration method is easier, faster, and has the big advantage of avoiding the use of a calibration grid. Instead of the calibration grid the measurement points themselves are used as the calibration points by determining their coordinates. A Leica surveillance device with a laser range finder is used in the calibration process to determine the exact position (x,y,z) of the calibration points. The markers are then used as 3D sensors of stereophotogrammetry after the calibration and the cameras record their motion to achieve the displacements by post processing of the images.

A comparison between the results from the new 3D calibration method and the traditional 2D calibration that is using a calibration grid shows a higher accuracy for the new 3D calibration procedure.

The new 3D calibration procedure was then conducted using different numbers of the calibration points on the turbine components and also on the background. It is concluded that the background points are not necessary and do not improve the calibration quality. This is very important in the full scale experiment as is would be problematic to install background points on a modern size wind turbine. The investigations also showed that decreasing the calibration points down to 35 points on

the turbine for an imaging area of 2 m×2 m, still gives acceptable quality on the 3D calibration.To measure the displacement of the markers during the rotation, a tracking algorithm has been developed based on the circular motion of the rotor markers and the robust corner detection image processing algorithms for determination of the makers' position. This algorithm that keeps tracking the markers robustly during operation, updates its parameters based on the angular deflection of the marker in the last time step and the elliptical path of the markers in the images during the rotation.

Light reflection and camera un-synchronization are discussed as the main sources of error during measurement that can be addressed using matt markers and also external trigger for the cameras in full scale experiments.

### Acknowledgments

The author would like to thank Per Hansen for his kind support in the experiments and also Rozenn Wagner for reviewing the paper and her valuable comments.

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
