# Peer review of "Establishing a robust testing approach for displacement measurement on a rotating horizontal axis wind turbine"

_Wind Energy Science, 2017_

## Referee Comment (RC1) · Anonymous Referee #1 · 10 Dec 2017

General comments

The study "Establishing a robust testing approach for displacement measurement on a rotating horizontal axis wind turbine" describes the experiment on a scaled model of operating wind turbine, where the blades displacements were measured using stereo photometry. The paper describes the calibration of the measurement system and the tracking procedure: the two important operations required when applying stereo photometry to operating wind turbines.

Though the paper contains important findings and recommendations, which could be quite useful for those who considers stereo photometry, paper's quality is not suffi-

ciently high to recommend it for publication. A major revision is necessary.

Specific comments

English requires some polishing: some of the paragraphs are not quite clear because of the language. Generally, the paper is written sloppy, there are many unexplained statements; sometimes, no details provided. Also the paper does not provide any critical assessment of the suggested techniques.

The paper claims "robust . . . measurement on a rotating . . . wind turbine", however only demonstrates the techniques on a quite small model and does not provide any considerations regarding the scalability of the measurement system.

The reference list could be updated: during the recent years quite many measurements campaigns were reported on real size operating wind turbines, and the references to campaigns from 2002 look quite outdated.

Quality of the figures needs significant improvement.

Technical corrections

P.2, line 23: should be "are", not "is"

P.2 line 32: using "well-defined" in this context is confusing.

P.3 line 14: why an Envision wind turbine is mentioned? Is it important in the context?

P.4 line 2. What is the "full resolution"?

P.4 lines 8-9. Consideration regarding the distances is very confusing. What do you mean?

P. 5, line 1. How the marker shown in Fig.3 helps to avoid the mentioned difficulties? Please explain.

P.6 line 17. Using term "smart device" is discussable in scientific literature. Please provide what is the functionality of the device. In the following text, usage of word "Leica" is too unceremonious. "Leica" is the name of a German company, which produces many other devices.

Fig. 5 seems to be rotated 90 degrees CCW. Why not to put it as it looks in reality? Same for fig.8.

P.7, line 9. Casualty: "the rotor rotates one cycle within 40 pictures" or "the camera takes 40 pictures during one rotor revolution"?

P.8 line 7. Where are "the first and second point (N1)"? It is unclear from fig.6.

Do the terms "line of sight" (p.9, line 6) and "light ray" (p.10, line 11) refer to the same? If yes, avoid using the both terms, if not, please explain the difference.

What does Table 1 mean? How do the numbers quantify the quality of calibration? Where is point 6? The part of the discussion regarding the calibration is very unclear and confusing and require thoughtful revising.

P.11 line 14. Is the "blade elongation" physical? I.e. the blades become longer due to the centrifugal forces? Please explain what do you mean here.

P.11 line 15. Where are the markers 1 and 2. If the marker numbers are important in the context, they should be shown in a figure.

P.12 line 1. The first paragraph: why this? Please provide more understandable explanation.

P.12 line 13: "by looking at pictures"... What pictures? Please provide figures' numbers.

Fig.12 needs numbering (e.g. a,b,c). The vertical axis of fig.12c, must be $U\_z$. Is it depth? What is the coordinate system? How the values on the graphs correlate the elongation values?

---

## Referee Comment (RC2) · Anonymous Referee #2 · 3 Jan 2018

Title: "Establishing a robust testing approach for displacement measurement on a rotating horizontal axis wind turbine" Paper No: wes-2017-49 Authors: Nadia Najafi, Allan Vesth

Dear Editors:

I would like to express that the current manuscript is suitable for publication. There are few minor corrections that must be addressed by the corresponding author: 1) I suggest, that the authors should mention the assumptions and/or limitations of the experimental setup. Could the experimental results be affected by using different materials with less or more flexibility or rigidity? 1) pp. 4, line 7: The selection of the

distance of 7.5m between the camera and the turbine was a result of a dimensional analysis? How did you come up with this particular distance? 2) In section 4, pp.7-8, I recommend add a flow chart of the tracking algorithm. Is it an open loop or closed loop algorithm? 3) In figure 12, reduce the scale for the y-axis as follows: Ux from 0.08-0.11, Uy from 0.08-0.11 and the mean value from 0.2 to 0.5 4) I suggest that the conclusion should be focused on the most relevant results obtained in the experiments. Details about the experimental setup and procedure should be mentioned in the experimental setup section (pp. 3). Is it realistic to apply the proposed 3D technique to a 3.6 wind turbine? 5) In pp. 14, line 16, the imaging area is supposed to be 2mx1.9m (see pp. 5, line 21) Regards, The reviewer

---

## Referee Comment (RC3) · Anonymous Referee #3 · 4 Jan 2018

**General Comments**

This work aims at developing a new 3D photogrammetric calibration technique which can be used for infield vibration tests on utility-scale horizontal axis wind turbines. For this purpose some dynamic experiments and measurements are conducted on a small scale wind turbine model in the lab.

Regarding the content, the methods described in the article can be very useful for the lab tests but unfortunately they are not applicable to the infield tests efficiently as claimed. Therefore, the paper needs significant major revisions in order to show that the proposed system can also be used in the field. Below you can see my recommendations

1- The term photometry is mis-used in the text. I think this word should be corrected as photogrammetry. Photometry: The science of measurement of visible light in terms of its perceived brightness to human vision. Photogrammetry: Determination of the 3D coordinates of the points on an object by using 2D images taken from different locations and orientations.

2- Photogrammetry can be easily used in small scale lab measurements performed in the controllable environments but infield tests have their own specific problems. For example, the authors mention about the accuracy of the device they used (Leica Nova MS50). The proposed accuracy is 0.035 mm in x and y directions and 1mm in z direction. However, this accuracy can never be reached in the field. It is not related to the accuracy of the device. In the field, the target will never be at standstill, it will be vibrating continuously. Even at low wind speeds the vibration amplitude can be +/- 10-20 cm. Besides, due to the mean wind speed, this vibration will not be a zero average vibration. How can you claim that you will reach 1 mm accuracy if the target itself is vibrating +/- 10-20 cm.

3- Leica Nova MS50 or similar total stations can only take measurement at one point at a time and then they move to the next data point. Therefore, it takes quite a lot of time to take measurements on 35 reference points. Could you please make an estimation related to time required to take measurements at 35 or maybe 100+ reference points? How can you guarantee that the wind speed so the vibration will not change within this period?

4- During the calibration at the site, how are you going to rotate the blades manually by a specific angle step by step? How are you going to move the device from one point to another and how are you going to guarantee that the physical conditions, blade pitch angle or yaw angle or wind speed (so the amplitude of the vibration) will be constant. These conditions can easily be fulfilled in a lab environment but not at the field. Under
these circumstances bundle adjustment method is the only possibility because you take measurements at all the points simultaneously. The method you proposed may provide a higher accuracy in lab environment where calibration is performed in an isolated room where there is no wind but not in the field where there is always some sort of wind and noisy vibration.

5- The extrinsic calibration values are valid only for a certain yaw and pitch angle. During the rotation these values do change continuously, and then you will have to recalibrate the system by using the new values. Are you planning to stop the turbine and to take some new calibration measurements with Leica system? You should explain in more detail how this method will be applied to the in-field tests?

6- Illumination is always the most important problem. That is why the markers you propose can only be used for close range of photogrammetry. For long range measurements, using reflective markers is the only choiceto reach the sufficient contrast levels. In the text page 14 line 20 you wrote that matt markers should be used. This suggestion makes the situation even worse, for long range measurements the markers should be as bright as possible to increase the contrast, otherwise the markers cannot be seen from long distances.

7- Page 2 line 5 "the transducers load the structure with their weight that changes the dynamic properties of the structure and need expensive correction". This statement is correct only if you perform some tests on very small models. I agree that an accelerometer of 100 grams can be considered as an added mass for a small scale model but weight of a real wind turbine or a real bridge is not affected by the weight of an accelerometer of 100 grams. Could you please remove this sentence?

8- P2 line 23 "is" should be "are"

- 9- P8 line 5 please change "angel" to "angle"
- 10- Page 11 line 13: You wrote that rotational speed of 30 rpm can cause an elongation
on metal rod blades. Could you please check these values again? I am not sure but I do not think that such a low speed can cause a noticeable elongation on metal rods?

11- Page 16: It is not clear how to read and interpret Table 1. Could you please explain in more detail what the distance between the light rays is? A sketch would be very helpful.

---

## Author Comment (AC1) · 11 Feb 2018

General comments The study "Establishing a robust testing approach for displacement measurement on a rotating horizontal axis wind turbine" describes the experiment on a scaled model of operating wind turbine, where the blades displacements were measured using stereo photometry. The paper describes the calibration of the measurement system and the tracking procedure: the two important operations required when applying stereo photometry to operating wind turbines. Though the paper contains important findings and recommendations, which could be quite useful for those who

considers stereo photometry, paper's quality is not sufficiently high to recommend it for publication. A major revision is necessary.

RESPONSE: Thank you for reconsidering the paper and constructive comments. In the following I tried to be precise in answering and satisfying the points. * All the comments are applied to the marked document that has been attached to the supplement. ** All the page numbers, reported in the answers, are based on the marked document that has been attached to the supplement.

Specific comments
* * *
COMMENT 1: English requires some polishing: some of the paragraphs are not quite clear because of the language. Generally, the paper is written sloppy, there are many unexplained statements; sometimes, no details provided. Also the paper does not provide any critical assessment of the suggested techniques. The text has been reworked and a discussions regarding challenges with the method's has been added. The paper claims "robust . . . measurement on a rotating . . . wind turbine", however only demonstrates the techniques on a quite small model and does not provide any considerations regarding the scalability of the measurement system.

RESPONSE: Considerations regarding the scalability of the measurement system have been added to page 15, lines 11-25.
* * *
COMMENT 2: The reference list could be updated: during the recent years quite many measurements campaigns were reported on real size operating wind turbines, and the references to campaigns from 2002 look quite outdated.

RESPONSE: The reference list is updated: In page 2, line 2, I replaced two reference from 2002 with two new works in the same filed from 2017 and 2014 and also added a new reference from 2016 to be an example of damage detection via traditional

transducers: - Weijtjens, W., Verbelen, T., Capello, E., Devriendt, C. (2017): Vibration based structural health monitoring of the substructures of five offshore wind turbines, pp. 2294-2299, Procedia Engineering 199. - Manzato, S., Santos, F., Peeters, B., LeBlanc, B., White, J. R. (2014): Combined accelerometers-strain gauges Operational Modal Analysis and application to wind turbine data, Proceedings of the 9th International Conference on Structural Dynamics: June 30-July 2, 2014, Porto, Portugal.

- Lorenzo, E. D., Petrone, G., Manzato, S., Peeters, B., Desmet, W., Marulo, F. (2016): Damage detection in wind turbine blades by using operational modal analysis, pp. 289-301, Structural Health Monitoring. 15(3). I also provided 3 new references from 2012, 2014 and 2015 to page 2, line 4 to refer to the measurement on large structures and bridges. - Ye, X. W., Ni, Ye. Q., Wong, K. Y., Ko, J. M. (2012): Statistical analysis of stress spectra for fatigue life assessment of steel bridges with structural health monitoring data, pp. 166–176, Engineering Structures 45.

- Xia, Z., Zhang, P., Ni, Y., Zhu, H. (2014): Deformation monitoring of a super-tall structure using real-time strain data, pp. 29–38, Engineering Structures 67.

- Siriwardane, S. C. (2015): Vibration measurement-based simple technique for damage detection of truss bridges: A case study, pp. 50–58, Case Studies in Engineering Failure Analysis 4.

\*\*\*\*\*\*\*\*\*\*\*\*\*\*\*\*\*\*\*\*\*\*\*\*\*\*\*\*\*\*\*\*\*\*

COMMENT 3: Quality of the figures needs significant improvement.

RESPONSE: The quality of Figures 1 and 5 has been improved. Figures 10-13 are also vector graphics and have a very high resolution.

\*\*\*\*\*\*\*\*\*\*\*\*\*\*\*\*\*\*\*\*\*\*\*\*\*\*\*\*\*\*\*\*\*\*

Technical corrections

COMMENT 4: P.2, line 23: should be "are", not "is".

RESPONSE: It is corrected.

\*\*\*\*\*\*\*\*\*\*\*\*\*\*\*\*\*\*\*\*\*\*\*\*\*\*\*\*\*\*\*\*\*\*\*\*\*\*\*\*

COMMENT 5: P.2 line 32: using "well-defined" in this context is confusing.

RESPONSE: I changed the sentence to avoid the confusion: "This study is focused on establishing a well-defined clearly described and easily applicable procedure to measure displacement on the components of a rotating horizontal axis wind turbine using stereo vision technique."

\*\*\*\*\*\*\*\*\*\*\*\*\*\*\*\*\*\*\*\*\*\*\*\*\*\*\*\*\*\*\*\*\*\*\*\*\*\*\*\*\*\*\*\*\*\*\*\*\*\*\*\*\*\*\*\*\*\*\*\*\*\*

COMMENT 6: P.3 line 14: why an Envision wind turbine is mentioned? Is it important in the context?

RESPONSE: It is mentioned to provide enough information about the case study which is a scaled down Envision wind turbine. However the Envision part is removed to avoid confusion.

\*\*\*\*\*\*\*\*\*\*\*\*\*\*\*\*\*\*\*\*\*\*\*\*\*\*\*\*\*\*\*\*\*\*\*\*\*\*\*\*\*\*\*\*\*\*\*\*\*\*\*\*\*\*\*\*\*\*\*\*\*\*

COMMENT 7: P.4 line 2. What is the "full resolution"?

RESPNOSE: It might not be the proper phrase in this sentence therefore I changed it to "full size".

\*\*\*\*\*\*\*\*\*\*\*\*\*\*\*\*\*\*\*\*\*\*\*\*\*\*\*\*\*\*\*\*\*\*\*\*\*\*\*\*\*\*\*\*\*\*\*\*\*\*\*\*\*\*\*\*\*\*\*\*\*\*

COMMENT 8: P.4 lines 8-9. Consideration regarding the distances is very confusing. What do you mean?

RESPONSE: The sentence is rephrased to give a better understanding: This setup satisfies the rule of thumb which says the distance between the cameras should be at least 1/3 of the distance between the cameras and the test object; it can be up to 3 times of the distance as long as all the targets on the object can be seen in the stereo

image pairs.

\*\*\*\*\*\*\*\*\*\*\*\*\*\*\*\*\*\*\*\*\*\*\*\*\*\*\*\*\*\*\*\*\*\*\*\*\*\*\*\*\*\*\*\*\*\*\*\*\*\*\*\*\*\*\*\*\*\*

COMMENT 9: P. 5, line 1. How the marker shown in Fig.3 helps to avoid the mentioned difficulties? Please explain.

RESPONSE: The paragraph is revised and more explanation is added to that.

\*\*\*\*\*\*\*\*\*\*\*\*\*\*\*\*\*\*\*\*\*\*\*\*\*\*\*\*\*\*\*\*\*\*\*\*\*\*\*\*\*\*\*\*\*\*\*\*\*\*\*\*\*\*\*\*\*\*\*

COMMENT 10: P.6 line 17. Using term "smart device" is discussable in scientific literature. Please provide what is the functionality of the device. In the following text, usage of word "Leica" is too unceremonious. "Leica" is the name of a German company, which produces many other devices.

RESPONSE: I agree with you comment, thus I removed the term "smart device" and added more explanation to the paragraph to describe Leica Nova MS50 with more details.

\*\*\*\*\*\*\*\*\*\*\*\*\*\*\*\*\*\*\*\*\*\*\*\*\*\*\*\*\*\*\*\*\*\*\*\*\*\*\*\*\*\*\*\*\*\*\*\*\*\*\*\*\*\*\*\*\*\*\*\*\*

COMMENT 11: Fig. 5 seems to be rotated 90 degrees CCW. Why not to put it as it looks in reality? Same for fig.8.

RESPONSE: The orientation of the images is because of the camera positioning on the camera holder. To eliminate the confusion, I rotate the figures 4, 5 & 8 to be in the same orientation of the real world.

\*\*\*\*\*\*\*\*\*\*\*\*\*\*\*\*\*\*\*\*\*\*\*\*\*\*\*\*\*\*\*\*\*\*\*\*\*\*\*\*\*\*\*\*\*\*\*\*\*\*\*\*\*\*\*\*\*\*\*\*\*

COMMENT 12: P.7, line 9. Casualty: "the rotor rotates one cycle within 40 pictures" or "the camera takes 40 pictures during one rotor revolution"?

RESPONSE: I think they both have the same meaning, but the sentence that you mentioned might be more clear, therefore I replace sentence "the rotor rotates one

cycle within 40 pictures" with the sentenced that you suggested: "the camera takes 40 pictures during one rotor revolution".
* * *
COMMENT 13: P.8 line 7. Where are "the first and second point (N1)"? It is unclear from fig.6.

RESPONSE: I added the position of the first and second points to Figure 6. In addition N1 is "the number of image sequences between the first and second position of the marker" as it is explained in page 8, line 7 and also in the caption of Figure 6.
* * *
COMMENT 14: Do the terms "line of sight" (p.9, line 6) and "light ray" (p.10, line 11) refer to the same? If yes, avoid using the both terms, if not, please explain the difference.

RESPONSE: The comment is absolutely true, they refer to the same thing, thus I revised the text and used from line of sight in the document.
* * *
COMMENT 15: What does Table 1 mean? How do the numbers quantify the quality of calibration?

RESPONSE: Table 1 presents the distance between the lines of sight during rotation for one of the markers and for different calibrations. The lines of sight from the marker to the cameras do not exactly intersect in the space due to the inaccuracies and the 3D position is regarded as the point with minimum distance from two lines of sight (Trucco and Verri, 1998). Therefore the distance between the lines of sight is considered as an indication of measurement inaccuracy that is mainly caused by calibration uncertainties and light reflections and also by other environmental and physical factors. The light and other environmental factors are almost the same in all the measurements in Table

1, therefore the different values of lines of sight distances are mostly due to different calibrations. For more clarification, more explanation is added to 3 paragraphs before Table 1.

\*\*\*\*\*\*\*\*\*\*\*\*\*\*\*\*\*\*\*\*\*\*\*\*\*\*\*\*\*\*\*\*\*\*\*\*\*\*\*\*\*\*\*\*\*\*\*\*\*\*\*\*\*\*\*\*\*\*\*\*\*\*\*\*\*

COMMENT 16: Where is point 6? The part of the discussion regarding the calibration is very unclear and confusing and require thoughtful revising.

RESPONSE: The point number is written by mistake. The correct form is "point 2 on the blade 2" (numbering of the markers is presented in Figure 8). It is also corrected in the text. I revised and also provided more explanation (to the paragraphs before Table 1) in this section.

\*\*\*\*\*\*\*\*\*\*\*\*\*\*\*\*\*\*\*\*\*\*\*\*\*\*\*\*\*\*\*\*\*\*\*\*\*\*\*\*\*\*\*\*\*\*\*\*\*\*\*\*\*\*\*\*\*\*\*\*\*\*\*\*\*

COMMENT 17: P.11 line 14. Is the "blade elongation" physical? I.e. the blades become longer due to the centrifugal forces? Please explain what do you mean here.

RESPONSE: This paragraph is added to page 10: "The distance between the markers will barely change during the turbine operation due to the centrifugal forces and gravity, thus the change of the distance between the markers when the turbine is rotating can be used as another indication of the measurement inaccuracy."

\*\*\*\*\*\*\*\*\*\*\*\*\*\*\*\*\*\*\*\*\*\*\*\*\*\*\*\*\*\*\*\*\*\*\*\*\*\*\*\*\*\*\*\*\*\*\*\*\*\*\*\*\*\*\*\*\*\*\*\*\*\*\*\*\*\*\*

COMMENT 18: P.11 line 15. Where are the markers 1 and 2. If the marker numbers are important in the context, they should be shown in a figure.

RESPONSE: The numbering of the markers is shown in Figure 8. For more clarification, Figure 8 is referred in page 11, line 18.

\*\*\*\*\*\*\*\*\*\*\*\*\*\*\*\*\*\*\*\*\*\*\*\*\*\*\*\*\*\*\*\*\*\*\*\*\*\*\*\*\*\*\*\*\*\*\*\*\*\*\*\*\*\*\*\*\*\*

COMMENT 19: P.12 line 1. The first paragraph: why this? Please provide more
understandable explanation.

RESPONSE: I agree that more information is needed, therefore more detailed explanation about blade elongation is added to page 10 (paragraphs before Table 1).

\*\*\*\*\*\*\*\*\*\*\*\*\*\*\*\*\*\*\*\*\*\*\*\*\*\*\*\*\*\*\*\*\*\*\*\*\*\*\*\*\*\*\*\*\*\*\*\*\*\*\*\*\*\*\*\*\*\*\*\*\*\*\*\*\*\*

COMMENT 20: P.12 line 13: "by looking at pictures"... What pictures? Please provide figures' numbers. The pictures that are taken and processed for the displacement measurement. This explanation is added to page 12 to avoid confusion.

\*\*\*\*\*\*\*\*\*\*\*\*\*\*\*\*\*\*\*\*\*\*\*\*\*\*\*\*\*\*\*\*\*\*\*\*\*\*\*\*\*\*\*\*\*\*\*\*\*\*\*\*\*\*\*\*\*\*\*\*\*\*\*\*\*\*

COMMENT 21: Fig.12 needs numbering (e.g. a,b,c). The vertical axis of fig.12c, must be U_z. Is it depth? What is the coordinate system? How the values on the graphs correlate the elongation values?

RESPONSE: Absolutely true, the numbering is added to Figure 12 and the vertical axis of fig.12c is changed to U_z which represents the uncertainty in depth. The coordinate system is shown in 5. Elongation values are introduced as an indication of the displacement measurements in this paper and are used to compare the accuracy of different calibration methods but uncertainty analysis quantifies the uncertainty of the measured displacement that is carried out with new proposed calibration method. Elongation value don't represent an specific coordinate but the uncertainty analysis using based on the law of error propagation represents the uncertainty values in x, y (in-plane) and z (out of plane) coordinates.

Please also note the supplement to this comment:
https://www.wind-energ-sci-discuss.net/wes-2017-49/wes-2017-49-AC1-supplement.pdf

---

## Author Comment (AC2) · 11 Feb 2018

Anonymous Referee #2 Received and published: 3 January 2018

Title: "Establishing a robust testing approach for displacement measurement on a rotating horizontal axis wind turbine" Paper No: wes-2017-49 Authors: Nadia Najafi, Allan Vesth

Dear Editors: I would like to express that the current manuscript is suitable for publication. There are few minor corrections that must be addressed by the corresponding author:

Thank you for reconsidering the paper and constructive comments. In the following I tried to be precise in answering and satisfying the points.

\* All the commentsare applied to the marked document that has been attached to the supplement.

\*\* Allthe page numbers, reported in the answers, are based on the marked document that has been attached to the supplement.

\*\*\*\*\*\*\*

COMMENT 1: I suggest, that the authors should mention the assumptions and/or limitations of the experimental setup. Could the experimental results be affected by using different materials with less or more flexibility or rigidity?

RESPONSE: It is a fair point. One of the assumptions is considering that the cameras follow the pinhole camera model that is added in page 4, second paragraph. The other limitation of the experiment is about camera synchronization that has been explained in page 14, second paragraph: "In the current study there is no external trigger or switch to start the cameras and they are triggered at the same time using software trigger (LabVIEW code) that could disturb the perfect synchronization between the cameras." There are also other limitations in the setup regarding the turbine yaw and pitch that have been explained in paragraph 3 and 4 in page 14.

The proposed method for displacement measurement is not affected by the flexibility or rigidity but if the material is too flexible the normal distance between two markers would change due to the rotation and the change of this parameter during rotation cannot be used as an indication of measurement inaccuracy.

COMMENT 2: pp. 4, line 7: The selection of the distance of 7.5m between the camera and the turbine was a result of a dimensional analysis? How did you come up with this particular distance?

WESD
RESPONSE: According to the turbine dimensions the imaging area is chosen to be almost 2mx2m, the lens focal length is 40 mm and the camera sensor size is 11.26 mm x 11.26 mm. Based on these factors and pinhole camera model, the distance between the cameras and turbine is calculated to be 7.5 m.

COMMENT 3:

In section 4, pp.7-8, I recommend add a flow chart of the tracking algorithm.

RESPONSE: Is it an open loop or closed loop algorithm? I added a flow chart at the end of the tracking algorithm section. It is a closed loop.

\*\*\*\*\*\*\*\*\*\*\*\*\*\*

COMMENT 4: In figure 12, reduce the scale for the y-axis as follows: Ux from 0.08-0.11, Uy from 0.08-0.11 and the mean value from 0.2 to 0.5.

RRESPONSE: Ux exceeds a bit from 0.11 so I set the range for Ux and Uy to 0.08-0.11 and for Uz to 0.2-0.5.

\*\*\*\*\*\*\*\*\*\*\*\*\*\*

COMMENT 5: I suggest that the conclusion should be focused on the most relevant results obtained in the experiments. Details about the experimental setup and procedure should be mentioned in the experimental setup section (pp. 3). Is it realistic to apply the proposed 3D technique to a 3.6 wind turbine?

RESPONSE: It is a fair point; I removed the part about the setup and procedure in the conclusion. With considering some practical points the technique can be applied on a full size wind turbine. New information and explanation is provided in page 15 about applying the proposed technique in filed.

**WESD**
COMMENT 6: In pp. 14, line 16, the imaging area is supposed to be 2mx1.9m (see pp. 5, line 21).

RESPONSE: 2mx1.9m is the dimensions of the 2D calibration grid while the imaging area is a bit bigger (about 2mx2m) as it can be seen in Figure 4.

Regards, The reviewer

Please also note the supplement to this comment: https://www.wind-energ-sci-discuss.net/wes-2017-49/wes-2017-49-AC2supplement.pdf

---

## Author Comment (AC3) · 11 Feb 2018

General Comments This work aims at developing a new 3D photogrammetric calibration technique which can be used for infield vibration tests on utility-scale horizontal axis wind turbines. For this purpose some dynamic experiments and measurements are conducted on a small scale wind turbine model in the lab. Regarding the content, the methods described in the article can be very useful for the lab tests but unfortu-

nately they are not applicable to the infield tests efficiently as claimed. Therefore, the paper needs significant major revisions in order to show that the proposed system can also be used in the field. Below you can see my recommendations

RESPONSE: Thank you for reconsidering the paper and constructive comments. In the following I tried to be precise in answering and satisfying the points.

* All the commentsare applied to the marked document that has been attached to the supplement.

** Allthe page numbers, reported in the answers, are based on the marked document that has been attached to the supplement.

\*\*\*\*\*\*\*\*\*\*\*\*\*\*\*\*\*\*\*\*\*\*\*\*\*\*\*\*\*\*\*\*\*\*\*\*\*\*\*\*\*\*\*\*\*\*\*\*\*\*\*\*\*\*\*\*\*\*\*\*\*\*

COMMENT 1: The term photometry is mis-used in the text. I think this word should be corrected as photogrammetry. Photometry: The science of measurement of visible light in terms of its perceived brightness to human vision. Photogrammetry: Determination of the 3D coordinates of the points on an object by using 2D images taken from different locations and orientations.

RESPONSE: The comment is absolutely true. I replaced photometry with photogrammetry in the whole document.

\*\*\*\*\*\*\*\*\*\*\*\*\*\*\*\*\*\*\*\*\*\*\*\*\*\*\*\*\*\*\*\*\*\*\*\*\*\*\*\*\*\*\*\*\*\*\*\*\*\*\*\*\*\*\*\*\*\*\*\*\*\*\*\*

COMMENT 2: Photogrammetry can be easily used in small scale lab measurements performed in the controllable environments but infield tests have their own specific problems. For example, the authors mention about the accuracy of the device they used (Leica Nova MS50). The proposed accuracy is 0.035 mm in x and y directions and 1mm in z direction. However, this accuracy can never be reached in the field. It is not related to the accuracy of the device. In the field, the target will never be at standstill, it will be vibrating continuously. Even at low wind speeds the vibration amplitude can be +/- 10- 20 cm. Besides, due to the mean wind speed, this vibration

will not be a zero average vibration. How can you claim that you will reach 1 mm accuracy if the target itself is vibrating +/- 10-20 cm.

RESPONSE: I totally agree that the experiment in the field is much more challenging thaen the experiment in the lab and one of the challenges will be definitely the vibrations during the calibration. Therefore the turbine vibrations should be considered during the calibration process and also uncertainty analysis. It could be suggested that the calibration is to be done in a calm weather condition with low wind to minimize the inaccuracies due to the wind induced vibrations. However the Leica Nova MS50 coordinate measurement is based on wave form digitizing technology (WFD) that is able to send out the short pulses with a frequency of up to 2 MHz toward the target [1] and it enables the Leica Nova MS50 to capture even very fast vibrations while the most dominant natural frequencies of the large wind turbines barley exceed 10 Hz. To notify the challenges that need to be considered in the field measurement, more explanation has been added to page 15 that provides information in paragraphs 3 and 4.

[1] Maar, H., Zogg, H. M. (2014): WFD – Wave Form Digitizer Technology Leica Geosystems AG Heerbrugg, Switzerland

\*\*\*\*\*\*\*\*\*\*\*\*\*\*\*\*\*\*\*\*\*\*\*\*\*\*\*\*\*\*\*\*\*\*\*\*\*\*\*\*\*\*\*\*\*\*\*\*\*\*\*\*\*\*

COMMENT 3: Leica Nova MS50 or similar total stations can only take measurement at one point at a time and then they move to the next data point. Therefore, it takes quite a lot of time to take measurements on 35 reference points. Could you please make an estimation related to time required to take measurements at 35 or maybe 100+ reference points? How can you guarantee that the wind speed so the vibration will not change within this period? Regarding the personal experience about Leica Nova MS50 coordinate measurement in the field, it takes less than 2 hours for the device to pick the coordinates of 100+ reference points.

RESPONSE: The changes of vibration and wind speed should be considered during the calibration, as you noted. This issue could be addressed with synchronizing the image acquisition system and Leica Leica surveillance device during camera calibration. This point is also mentioned in the added information about experiment in the field in page 14.

\*\*\*\*\*\*\*\*\*\*\*\*\*\*\*\*\*\*\*\*\*\*\*\*\*\*\*\*\*\*\*\*\*\*\*\*\*\*\*\*\*\*\*\*\*\*\*\*\*\*\*\*\*\*\*\*\*\*\*\*\*\*\*\*\*\*\*

COMMENT 4: During the calibration at the site, how are you going to rotate the blades manually by a specific angle step by step? How are you going to move the device from one point to another and how are you going to guarantee that the physical conditions, blade pitch angle or yaw angle or wind speed (so the amplitude of the vibration) will be constant. These conditions can easily be fulfilled in a lab environment but not at the field. Under these circumstances bundle adjustment method is the only possibility because you take measurements at all the points simultaneously. The method you proposed may provide a higher accuracy in lab environment where calibration is performed in an isolated room where there is no wind but not in the field where there is always some sort of wind and noisy vibration.

RESPONSE: They are fair points and should be considered during full scale experiment. During the camera calibration in the field, we should break the turbine and then release the break shortly until it reaches the expected position. The pitch and yaw angles could also be locked within the calibration and the vibration amplitude and wind speed change can be addressed using the Leica surveillance device and camera synchronization.

\*\*\*\*\*\*\*\*\*\*\*\*\*\*\*\*\*\*\*\*\*\*\*\*\*\*\*\*\*\*\*\*\*\*\*\*\*\*\*\*\*\*\*\*\*\*\*\*\*\*\*\*\*\*\*\*\*\*\*\*\*\*\*\*\*

COMMENT 5: The extrinsic calibration values are valid only for a certain yaw and pitch angle. During the rotation these values do change continuously, and then you will have to recalibrate the system by using the new values. Are you planning to stop the turbine and to take some new calibration measurements with Leica system? You should explain in more detail how this method will be applied to the in-field tests?

RESPONSE: The calibration method that is proposed in this paper is a 3D calibration due to the 3D distribution of the points on the rotor, nacelle and turbine tower. Therefore if the turbine yaws or the blades pitch within a reasonable range, the calibration is still valid for the measurement and there is no need for recalibration. However I absolutely agree that a comprehensive study should be done on the range where camera calibration is valid in the filed measurement.
* * *
COMMENT 6: Illumination is always the most important problem. That is why the markers you propose can only be used for close range of photogrammetry. For long range measurements, using reflective markers is the only choiceto reach the sufficient contrast levels. In the text page 14 line 20 you wrote that matt markers should be used. This suggestion makes the situation even worse, for long range measurements the markers should be as bright as possible to increase the contrast, otherwise the markers cannot be seen from long distances.

RESPONSE: Illumination and reflection are one of the main challenges in photogrammetry, as you mentioned. Using reflective markers is associated with some problems such as not enough contrast with the background during the day time, providing sufficient illumination for the markers on large structures, significant changes of reflection angles and reflection quality during turbine operation and also miscalculation of the rotational plan that cause systematic errors during the measurement. Non-reflective, black and white markers have been already used for displacement measurements on large structures such as bridges [1,2,3] that proves that optical long range measurements are possible using non-reflective markers and with choosing the proper size and shape for the markers. In addition they do not need extra light sources than the sun (day light) and with having matt markers even the day light reflection on the markers is avoided to a great extent that decreases a lot the systematic error due to the light reflection. [1] Lee, J. J., and Shinozuka, M. (2006): Real-Time Displacement Measurement of a Flexible Bridge Using Digital Image Processing Techniques, pp.105-114,

Experimental Mechanics (46). [2] Feng, D., Feng, M. Q., Ozer, E., and Fukuda, Y. (2015): A Vision-Based Sensor for Noncontact Structural Displacement Measurement, pp.16557-16575, Sensors (15). [3] Yang, J., Peng, C., Xiao, J., Zeng, J., and Yuan, Y. (2012): Application of videometric technique to deformation measurement for large-scale composite wind turbine blade, pp.292-300, Applied Energy (98).

\*\*\*\*\*\*\*\*\*\*\*\*\*\*\*\*\*\*\*\*\*\*\*\*\*\*\*\*\*\*\*\*\*\*\*\*\*\*\*\*\*\*\*\*\*\*\*\*\*\*\*\*\*\*\*\*\*\*\*\*

COMMENT 7: Page 2 line 5 ''the transducers load the structure with their weight that changes the dynamic properties of the structure and need expensive correction''. This statement is correct only if you perform some tests on very small models. I agree that an accelerometer of 100 grams can be considered as an added mass for a small scale model but weight of a real wind turbine or a real bridge is not affected by the weight of an accelerometer of 100 grams. Could you please remove this sentence?

RESPONSE: The sentence is removed.

\*\*\*\*\*\*\*\*\*\*\*\*\*\*\*\*\*\*\*\*\*\*\*\*\*\*\*\*\*\*\*\*\*\*\*\*\*\*\*\*\*\*\*\*\*\*\*\*\*\*\*\*\*\*\*\*\*\*\*\*

COMMENT 8: P2 line 23 ''is'' should be ''are''.

RESPONSE: Fair point, it is corrected.

\*\*\*\*\*\*\*\*\*\*\*\*\*\*\*\*\*\*\*\*\*\*\*\*\*\*\*\*\*\*\*\*\*\*\*\*\*\*\*\*\*\*\*\*\*\*\*\*\*\*\*\*\*\*\*\*\*\*\*\* COMMENT 9: P8 line 5 please change ''angel to ''angle''.

RESPONSE: Fair point, it is corrected.

\*\*\*\*\*\*\*\*\*\*\*\*\*\*\*\*\*\*\*\*\*\*\*\*\*\*\*\*\*\*\*\*\*\*\*\*\*\*\*\*\*\*\*\*\*\*\*\*\*\*\*\*\*\*\*\*\*\*\*\*

COMMENT 10:

Page 11 line 13: You wrote that rotational speed of 30 rpm can cause an elongation on metal rod blades. Could you please check these values again? I am not sure but I do not think that such a low speed can cause a noticeable elongation on metal rods?

RESPONSE: That is true, this part is removed from the text.
* * *
COMMENT 11: Page 16: It is not clear how to read and interpret Table 1. Could you please explain in more detail what the distance between the light rays is? A sketch would be very helpful.

RESPONSE: More explanation regarding Table 1 is added to page 11, just before Table 1.

Please also note the supplement to this comment:
https://www.wind-energ-sci-discuss.net/wes-2017-49/wes-2017-49-AC3-
supplement.pdf

**Supplement:**

[revised manuscript text omitted]

---

## Author Response (AR1)

**General comments**

The study "Establishing a robust testing approach for displacement measurement on a rotating horizontal axis wind turbine" describes the experiment on a scaled model of operating wind turbine, where the blades displacements were measured using stereo photometry. The paper describes the calibration of the measurement system and the tracking procedure: the two important operations required when applying stereo photometry to operating wind turbines.

Though the paper contains important findings and recommendations, which could be quite useful for those who considers stereo photometry, paper's quality is not sufficiently high to recommend it for publication. A major revision is necessary.

Thank you for reconsidering the paper and constructive comments. In the following I tried to be precise in answering and satisfying the points.

\*\* All the page numbers, reported in the answers, are based on the latest marked manuscript at the end of the document (after the responses to the reviewers).

**Specific comments**

English requires some polishing: some of the paragraphs are not quite clear because of the language. Generally, the paper is written sloppy, there are many unexplained statements; sometimes, no details provided. Also the paper does not provide any critical assessment of the suggested techniques. The text has been reworked and a discussion regarding challenges with the method has been added.

The paper claims "robust . . . measurement on a rotating . . . wind turbine", however only demonstrates the techniques on a quite small model and does not provide any considerations regarding the scalability of the measurement system. Considerations regarding the scalability of the measurement system have been added.

The reference list could be updated: during the recent years quite many measurements campaigns were reported on real size operating wind turbines, and the references to campaigns from 2002 look quite outdated. The reference list is updated:

In page 2, line 2, I replaced two reference from 2002 with two new works in the same filed from 2017 and 2014 and also added a new reference from 2016 to be an example of damage detection via traditional transducers:

- Weijtjens, W., Verbelen, T., Capello, E., Devriendt, C. (2017): Vibration based structural health monitoring of the substructures of five offshore wind turbines, pp. 2294-2299, Procedia Engineering 199.
- Manzato, S., Santos, F., Peeters, B., LeBlanc, B., White, J. R. (2014): Combined accelerometers-strain gauges Operational Modal Analysis and application to wind turbine data, Proceedings of the 9th International Conference on Structural Dynamics: June 30-July 2, 2014, Porto, Portugal.
- Lorenzo, E. D., Petrone, G., Manzato, S., Peeters, B., Desmet, W., Marulo, F. (2016): Damage detection in wind turbine blades by using operational modal analysis, pp. 289-301, Structural Health Monitoring. 15(3).

I also provided 3 new references from 2012, 2014 and 2015 to page 2, line 4 to refere to the measurement on large structures and bridges.

- Ye, X. W., Ni, Ye. Q., Wong, K. Y., Ko, J. M. (2012): Statistical analysis of stress spectra for fatigue life assessment of steel bridges with structural health monitoring data, pp. 166–176, Engineering Structures 45.
- Xia, Z., Zhang, P., Ni, Y., Zhu, H. (2014): Deformation monitoring of a super-tall structure using realtime strain data, pp. 29–38, Engineering Structures 67.
- Siriwardane, S. C. (2015): Vibration measurement-based simple technique for damage detection of truss bridges: A case study, pp. 50–58, Case Studies in Engineering Failure Analysis 4.

Quality of the figures needs significant improvement. The quality of Figures 1 and 5 has been improved. Figures 10-13 are also vector graphics and have a very high resolution.

**Technical corrections**

P.2, line 23: should be "are", not "is". It is corrected.

P.2 line 32: using "well-defined" in this context is confusing. I changed the phrase in the whole document to avoid the confusion: "This study is focused on establishing a well-defined clearly described and easily

applicable procedure to measure displacement on the components of a rotating horizontal axis wind turbine using stereo vision technique."

P.3 line 14: why an Envision wind turbine is mentioned? Is it important in the context? It is mentioned to provide enough information about the case study which is a scaled down Envision wind turbine. However the Envision part is removed to avoid confusion.

P.4 line 2. What is the "full resolution"? It might not be the proper phrase in this sentence therefore I changed it to "full size".

P.4 lines 8-9. Consideration regarding the distances is very confusing. What do you mean? The sentence is rephrased to give a better understanding:

This setup satisfies the rule of thumb which says the distance between the cameras should be at least 1/3 of the distance between the cameras and the test object; it can be up to 3 times of the distance as long as all the targets on the object can be seen in the stereo image pairs.

P. 5, line 1. How the marker shown in Fig.3 helps to avoid the mentioned difficulties? Please explain. The paragraph is revised and more explanation is added to that.

P.6 line 17. Using term "smart device" is discussable in scientific literature. Please provide what is the functionality of the device. In the following text, usage of word "Leica" is too unceremonious. "Leica" is the name of a German company, which produces many other devices. I agree with you comment, thus I removed the term "smart device" and added more explanation to the paragraph to describe Leica Nova MS50 with more details.

Fig. 5 seems to be rotated 90 degrees CCW. Why not to put it as it looks in reality? Same for fig.8. The orientation of the images is because of the camera positioning on the camera holder. To eliminate the confusion, I rotate the figures 4, 5 & 8 to be in the same orientation of the real world.

P.7, line 9. Casualty: "the rotor rotates one cycle within 40 pictures" or "the camera takes 40 pictures during one rotor revolution"? I think they both have the same meaning, but the sentence that you mentioned might be more clear, therefore I replace sentence "the rotor rotates one cycle within 40

pictures" with the sentenced that you suggested: "the camera takes 40 pictures during one rotor revolution".

P.8 line 7. Where are "the first and second point (N1)"? It is unclear from fig.6. I added the position of the first and second points to Figure 6. In addition N1 is "the number of image sequences between the first and second position of the marker" as it is explained in page 8, line 7 and also in the caption of Figure 6.

Do the terms "line of sight" (p.9, line 6) and "light ray" (p.10, line 11) refer to the same? If yes, avoid using the both terms, if not, please explain the difference. The comment is absolutely true, they refer to the same thing, thus I revised the text and used from line of sight in the document.

What does Table 1 mean? How do the numbers quantify the quality of calibration? Table 1 presents the distance between the lines of sight during rotation for one of the markers and for different calibrations. The lines of sight from the marker to the cameras do not exactly intersect in the space due to the inaccuracies and the 3D position is regarded as the point with minimum distance from two lines of sight (Trucco and Verri, 1998). Therefore the distance between the lines of sight is considered as an indication of measurement inaccuracy that is mainly caused by calibration uncertainties and light reflections and also by other environmental and physical factors. The light and other environmental factors are almost the same in all the measurements in Table 1, therefore the different values of lines of sight distances are mostly due to different calibrations.

For more clarification, more explanation is added to 3 paragraphs before Table 1.

Where is point 6? The part of the discussion regarding the calibration is very unclear and confusing and require thoughtful revising. The point number is written by mistake. The correct form is "point 2 on the blade 2" (numbering of the markers is presented in Figure 8). It is also corrected in the text.

I revised and also provided more explanation (to the paragraphs before Table 1) in this section.

P.11 line 14. Is the "blade elongation" physical? I.e. the blades become longer due to the centrifugal forces? Please explain what do you mean here.. This paragraph is added to page 10:

"The distance between the markers will barely change during the turbine operation due to the centrifugal forces and gravity, thus the change of the distance between the markers when the turbine is rotating can be used as another indication of the measurement inaccuracy."

P.11 line 15. Where are the markers 1 and 2. If the marker numbers are important in the context, they should be shown in a figure. The numbering of the markers is shown in Figure 8. For more clarification, Figure 8 is referred in page 11, line 18.

P.12 line 1. The first paragraph: why this? Please provide more understandable explanation. I agree that more information is needed, therefore more detailed explanation about blade elongation is added to page 10 (paragraphs before Table 1).

P.12 line 13: "by looking at pictures"... What pictures? Please provide figures' numbers. The pictures that are taken and processed for the displacement measurement. This explanation is added to page 12 to avoid confusion.

Fig.12 needs numbering (e.g. a,b,c). The vertical axis of fig.12c, must be U\_z. Is it depth? What is the coordinate system? How the values on the graphs correlate the elongation values? Absolutely true, the numbering is added to Figure 12 and the vertical axis of fig.12c is changed to U\_z which represents the uncertainty in depth.

The coordinate system is shown in 5

Elongation values are introduced as an indication of the displacement measurements in this paper and are used to compare the accuracy of different calibration methods but uncertainty analysis quantifies the uncertainty of the measured displacement that is carried out with new proposed calibration method. Elongation value don't represent an specific coordinate but the uncertainty analysis using based on the law of error propagation represents the uncertainty values in x, y (in-plane) and z (out of plane) coordinates.

Title: "Establishing a robust testing approach for displacement measurement on a rotating horizontal axis wind turbine" Paper No: wes-2017-49 Authors: Nadia Najafi, Allan Vesth

**Dear Editors:**

I would like to express that the current manuscript is suitable for publication. There are few minor corrections that must be addressed by the corresponding author:

Thank you for reconsidering the paper and constructive comments. In the following I tried to be precise in answering and satisfying the points.

\*\* All the page numbers, reported in the answers, are based on the latest marked manuscript at the end of the document (after the responses to the reviewers).

1) I suggest, that the authors should mention the assumptions and/or limitations of the experimental setup. Could the experimental results be affected by using different materials with less or more flexibility or rigidity? It is a fair point. One of the assumptions is considering that the cameras follow the pinhole camera model that is added in page 4, second paragraph. The other limitation of the experiment is about camera synchronization that has been explained in page 14, second paragraph: "In the current study there is no external trigger or switch to start the cameras and they are triggered at the same time using software trigger (LabVIEW code) that could disturb the perfect synchronization between the cameras." There are also other limitations in the setup regarding the turbine yaw and pitch that have been explained in paragraph 3 and 4 in page 14.

The proposed method for displacement measurement is not affected by the flexibility or rigidity but if the material is too flexible the normal distance between two markers would change due to the rotation and the change of this parameter during rotation cannot be used as an indication of measurement inaccuracy.

1) pp. 4, line 7: The selection of the distance of 7.5m between the camera and the turbine was a result of a dimensional analysis? How did you come up with this particular distance? According to the turbine dimensions the imaging area is chosen to be almost 2mx2m, the lens focal length is 40 mm and the camera sensor size is 11.26 mm x 11.26 mm. Based on these factors and pinhole camera model, the distance between the cameras and turbine is calculated to be 7.5 m.

2) In section 4, pp.7-8, I recommend add a flow chart of the tracking algorithm. Is it an open loop or closed loop algorithm? I added a flow chart at the end of the tracking algorithm section.

It is a closed loop.

3) In figure 12, reduce the scale for the y-axis as follows: Ux from 0.08-0.11, Uy from 0.08-0.11 and the mean value from 0.2 to 0.5. Ux exceeds a bit from 0.11 so I set the range for Ux and Uy to 0.08-0.11 and for Uz to 0.2-0.5.

4) I suggest that the conclusion should be focused on the most relevant results obtained in the experiments. Details about the experimental setup and procedure should be mentioned in the experimental setup section (pp. 3). Is it realistic to apply the proposed 3D technique to a 3.6 wind turbine? It is a fair point; I removed the part about the setup and procedure in the conclusion.

With considering some practical points the technique can be applied on a full size wind turbine. New information and explanation is provided in page 15 about applying the proposed technique in filed.

5) In pp. 14, line 16, the imaging area is supposed to be 2mx1.9m (see pp. 5, line 21). 2mx1.9m is the dimensions of the 2D calibration grid while the imaging area is a bit bigger (about 2mx2m) as it can be seen in Figure 4.

Regards, The reviewer

This work aims at developing a new 3D photogrammetric calibration technique which can be used for infield vibration tests on utility-scale horizontal axis wind turbines. For this purpose some dynamic experiments and measurements are conducted on a small scale wind turbine model in the lab.

Regarding the content, the methods described in the article can be very useful for the lab tests but unfortunately they are not applicable to the infield tests efficiently as claimed. Therefore, the paper needs significant major revisions in order to show that the proposed system can also be used in the field. Below you can see my recommendations

Thank you for reconsidering the paper and constructive comments. In the following I tried to be precise in answering and satisfying the points.

\*\* All the page numbers, reported in the answers, are based on the latest marked manuscript at the end of the document (after the responses to the reviewers).

1- The term photometry is mis-used in the text. I think this word should be corrected as photogrammetry. Photometry: The science of measurement of visible light in terms of its perceived brightness to human vision. Photogrammetry: Determination of the 3D coordinates of the points on an object by using 2D images taken from different locations and orientations. The comment is absolutely true. I replaced photometry with photogrammetry in the whole document.

2- Photogrammetry can be easily used in small scale lab measurements performed in the controllable environments but infield tests have their own specific problems. For example, the authors mention about the accuracy of the device they used (Leica Nova MS50). The proposed accuracy is 0.035 mm in x and y directions and 1mm in z direction. However, this accuracy can never be reached in the field. It is not related to the accuracy of the device. In the field, the target will never be at standstill, it will be vibrating continuously. Even at low wind speeds the vibration amplitude can be +/- 10- 20 cm. Besides, due to the mean wind speed, this vibration will not be a zero average vibration. How can you claim that you will reach 1 mm accuracy if the target itself is vibrating +/- 10-20 cm. I agree that the experiment in the field is much more challenging than the experiment in the lab and one of the challenges will be the vibrations during the calibration. Therefore the turbine vibrations should be considered during the calibration process and also

uncertainty analysis. It could be suggested that the calibration is to be done in a calm weather condition with low wind to minimize the inaccuracies due to the wind induced vibrations. However the Leica Nova MS50 coordinate measurement is based on wave form digitizing technology (WFD) that is able to send out the short pulses with a frequency of up to 2 MHz toward the target [1] and it enables the Leica Nova MS50 to capture even very fast vibrations while the most dominant natural frequencies of the large wind turbines barley exceed 10 Hz.

To notify the challenges that need to be considered in the field measurement, more explanation has been added to page 15 that provides information in paragraphs 3 and 4.

[1] Maar, H., Zogg, H. M. (2014): WFD – Wave Form Digitizer Technology Leica Geosystems AG Heerbrugg, Switzerland

3- Leica Nova MS50 or similar total stations can only take measurement at one point at a time and then they move to the next data point. Therefore, it takes quite a lot of time to take measurements on 35 reference points. Could you please make an estimation related to time required to take measurements at 35 or maybe 100+ reference points? How can you guarantee that the wind speed so the vibration will not change within this period? Regarding the personal experience about Leica Nova MS50 coordinate measurement in the field, it takes less than 2 hours for the device to pick the coordinates of 100+ reference points.

The changes of vibration and wind speed should be considered during the calibration, as you noted. This issue could be addressed with synchronizing the image acquisition system and Leica Leica surveillance device during camera calibration. This point is also mentioned in the added information about experiment in the field in page 14.

4- During the calibration at the site, how are you going to rotate the blades manually by a specific angle step by step? How are you going to move the device from one point to another and how are you going to guarantee that the physical conditions, blade pitch angle or yaw angle or wind speed (so the amplitude of the vibration) will be constant. These conditions can easily be fulfilled in a lab environment but not at the field. Under these circumstances bundle adjustment method is the only possibility because you take measurements at all the points simultaneously. The method you proposed may provide a higher accuracy in lab environment where calibration is performed in an isolated room where there is no wind but not in the field where there is always some sort of wind and noisy vibration. They are fair points and should be considered during full scale experiment. During the camera calibration in the field, we should break the turbine and then release the break shortly until it reaches the expected position. The pitch and yaw angles could also be locked within the calibration and the vibration amplitude and wind speed change can be addressed using the Leica surveillance device and camera synchronization.

5- The extrinsic calibration values are valid only for a certain yaw and pitch angle. During the rotation these values do change continuously, and then you will have to recalibrate the system by using the new values.

Are you planning to stop the turbine and to take some new calibration measurements with Leica system? You should explain in more detail how this method will be applied to the in-field tests? The calibration method that is proposed in this paper is a 3D calibration due to the 3D distribution of the points on the rotor, nacelle and turbine tower. Therefore if the turbine yaws or the blades pitch within a reasonable range, the calibration is still valid for the measurement and there is no need for recalibration. However I absolutely agree that a comprehensive study should be done on the range where camera calibration is valid in the filed measurement.

6- Illumination is always the most important problem. That is why the markers you propose can only be used for close range of photogrammetry. For long range measurements, using reflective markers is the only choiceto reach the sufficient contrast levels. In the text page 14 line 20 you wrote that matt markers should be used. This suggestion makes the situation even worse, for long range measurements the markers should be as bright as possible to increase the contrast, otherwise the markers cannot be seen from long distances. Illumination and reflection are one of the main challenges in photogrammetry, as you mentioned. Using reflective markers is associated with some problems such as not enough contrast with the background during the day time, providing sufficient illumination for the markers on large structures, significant changes of reflection angles and reflection quality during turbine operation and also miscalculation of the rotational plan that cause systematic errors during the measurement. Non-reflective, black and white markers have been already used for displacement measurements on large structures such as bridges [1,2,3] that proves that optical long range measurements are possible using non-reflective markers and with choosing the proper size and shape for the markers. In addition they do not need extra light sources than the sun (day light) and with having matt markers even the day light reflection.

- [1] Lee, J. J., and Shinozuka, M. (2006): Real-Time Displacement Measurement of a Flexible Bridge Using Digital Image Processing Techniques, pp.105-114, Experimental Mechanics (46).
- [2] Feng, D., Feng, M. Q., Ozer, E., and Fukuda, Y. (2015): A Vision-Based Sensor for Noncontact Structural Displacement Measurement, pp.16557-16575, Sensors (15).
- [3] Yang, J., Peng, C., Xiao, J., Zeng, J., and Yuan, Y. (2012): Application of videometric technique to deformation measurement for large-scale composite wind turbine blade, pp.292-300, Applied Energy (98).

7- Page 2 line 5 "the transducers load the structure with their weight that changes the dynamic properties of the structure and need expensive correction". This statement is correct only if you perform some tests on very small models. I agree that an accelerometer of 100 grams can be considered as an added mass for a small scale model but weight of a real wind turbine or a real bridge is not affected by the weight of an accelerometer of 100 grams. Could you please remove this sentence? The sentence is removed.

8- P2 line 23 "is" should be "are". Fair point, it is corrected.

9- P8 line 5 please change "angel to "angle". Fair point, it is corrected.

10- Page 11 line 13: You wrote that rotational speed of 30 rpm can cause an elongation on metal rod blades. Could you please check these values again? I am not sure but I do not think that such a low speed can cause a noticeable elongation on metal rods? That is true, this part is removed from the text.

11- Page 16: It is not clear how to read and interpret Table 1. Could you please explain in more detail what the distance between the light rays is? A sketch would be very helpful. More explanation regarding Table 1 is added to page 11, just before Table 1.

[revised manuscript text omitted]

**3.2 3D updated camera calibration with a surveillance device with a laser range finder Leica**

- Traditional 3D calibration method can be conducted very accurately, as it has been referred at the beginning of this section,
  but it needs expensive and elaborate setup and expensive equipment such as two or three orthogonal planes. Setting up the traditional 3D calibration in large field of views, like full scale wind turbine experiment, is not feasible due to the difficulties in providing a precise calibration object. In our new calibration method, instead of using a huge 3D calibration object, we used the markers that are applied on the turbine for the sake of measurement, as the calibration points. A Leica surveillance device (Leica Nova MS50) is our smart device to determines the exact position of the points quite accurately and then the
  coordinates are used for 3D calibration of each camera. Leica Nova MS50, is a surveillance device with a laser range finder
- that uses advanced technologies for 3D laser scanning, imaging and GNSS (Global Navigation Satellite System) positioning. The Leica-accuracy in x and y directions is dependent on the distance between the Leica Nova MS50 and the object, hence the accuracy in x and y directions in the current case is about 0.0349mm. In addition the Leica Nova MS50 accuracy in depth (z direction) is 1mm for measuring on reflective surfaces.
- 20
  - During the calibration the rotor was rotated (manually) by a specific angle step by step, in order to cover the whole rotor area by the calibration points and establish the collection of known coordinates for camera calibration.